# Characterization and Antibacterial Properties of Egg White Protein Films Loaded with ε-Polylysine: Evaluation of Their Degradability and Application

**DOI:** 10.3390/foods12122431

**Published:** 2023-06-20

**Authors:** Xin Li, Jianhao Lv, Minghao Niu, Siqi Liu, Yue Wu, Jiahan Liu, Jingwen Xie, Chengfeng Sun, Yue-Meng Wang

**Affiliations:** 1School of Life Sciences, Yantai University, Yantai 264005, China; ljh17853509809@163.com (J.L.); nmh@ytu.edu.cn (M.N.); sageliu_xxx@163.com (S.L.); w2573630513@163.com (Y.W.); jhoney19980312@163.com (J.L.); 18763449912@163.com (J.X.); yanjunyangx@126.com (C.S.); 2School of Food and Biological Engineering, Yantai Institute of Technology, Yantai 264003, China

**Keywords:** bioactive film, antibacterial properties, ecologically friendly, application in chilled pork

## Abstract

There is an ongoing trend to design new kinds of food packaging materials with excellent properties which are environmentally friendly enough. The aim of this study was to prepare and characterize egg white protein (EWP)-based composite films with and without ε-polylysine (Lys), as well as to compare their physical–chemical properties, structural properties, degradation and antibacterial properties. The results showed that with the addition of Lys, the composite films showed a decreasing tendency of the water permeability due to the enhanced interaction between proteins and water molecules. As indicated by the structural properties, stronger cross-linking and intermolecular interactions happened with increasing concentration of Lys. In addition, the composite films presented excellent antibacterial activities against *Escherichia coli* and *Staphylococcus aureus* on chilled pork in the presence of Lys. Therefore, our prepared films might be used as a freshness-keeping material with an application in meat preservation. The biodegradation evaluation demonstrated that the composite films were environmental-friendly and have potential applications in the field of food packaging.

## 1. Introduction

Nowadays, plastics are widely used in various fields, such as agriculture, industry, construction, packaging, cutting-edge defense industries and people’s daily lives [1]. The widespread use of plastics has caused irreversible harm to the environment on which people rely for survival [2,3]. It is known that traditional plastics are mainly made of high-molecular-weight compounds such as polystyrene, polypropylene and polyvinyl chloride. The chemical structure of these polymer compounds is stable, making them difficult to degrade by microorganisms. It is reported that the degradation of traditional plastics in the natural environment takes 200 to 400 years [4,5]. Therefore, arbitrary disposal of traditional plastics is extremely prone to environmental pollution. In order to solve the problem, developing a biodegradable functional film has become a key research project in the field of food packaging [6,7].

Egg white protein (EWP) is a kind of full-price high-quality protein with a wide range of sources and has good gel properties which give EWP a unique advantage as a film-forming matrix [8]. However, the application of pure EWP films in actual packaging processes is greatly limited due to their disadvantages, such as poor water vapor permeability (WVP), easy nourishment of microorganisms, etc. [9]. Therefore, using EWP as a matrix and combining it with other polymers to improve the performance of EWP-based films has become a new research hotspot. Studies have shown that composite films prepared by mixing EWP with k-carrageenan exhibit a decrease in WVP compared to the EWP film without k-carrageenan [10]. However, the enhancement of functionality requires seeking an ideal additive for compounding. In the past few years, various bacteriostatic agents have been incorporated into biological macromolecular films to enhance their antibacterial properties as well as their practical applications [11,12,13,14,15]. Ma Yuan’s team studied and prepared a konjac glucomannan (KGM)/chitosan (CTS) antibacterial film with oregano essential oil. The analysis of antibacterial properties showed that the addition of microcapsules improved the antibacterial performance of the KGM/CTS film and showed a better freshness during the storage of cold fresh pork when packed with the composite film [16]. However, there are currently not many reports on improving the antibacterial properties of egg white protein films. The potential application of EWP-based films in food industries still needs further research. 

Polylysine (Lys) is an efficient, safe and environmentally friendly natural preservative. It is non-toxic and biodegradable and can be widely used for food preservation. Meanwhile, Lys can effectively inhibit the growth of spoilage bacteria, maintain the freshness of the product and, thus, extend the shelf life of the product. Ma, Yuan, Feng, Wang, Sun, Cao and Huang [17] reported that Lys can significantly ameliorate the gel-forming ability of meat proteins. In addition, Lys could be incorporated with carboxymethyl cellulose to increase the antibacterial activity of the composited films and could be a potential material for food packaging [18]. Therefore, both biodegradability and antibacterial properties could be guaranteed in biomacromolecule-based composite films by the addition of Lys.

The purpose of this experiment is to prepare EWP-Lys composite films and determine the physical and chemical properties, degradation ability and antibacterial effect of the films during the preservation of cold fresh meat and clarify their characteristics.

## 2. Materials and Methods

### 2.1. Chemicals

Egg white protein powder, which is greater than 95% purity, was kindly donated by Kangde Egg Industry Co., Ltd. (Nantong, China). Glycerin and ε-polylysine were purchased from Sinopharm Chemical Reagent Co., Ltd. Foodborne pathogens (*Escherichia coli* and *Staphylococcus aureus*) were kindly donated by the Bacterial Fungus Research Center in Yantai University. Pork tenderloin originated from New World Department Store (Yantai, China). 

### 2.2. Film Formation Procedure

The composite films were prepared based on the previous literature [19] using the solvent casting method. Briefly, the film-forming EWP solutions (8.0% protein concentration, *w*/*w*) prepared with egg white powder were mixed with different concentrations of Lys (0%, 0.4%, 0.8%, 1.2%, 1.6% and 2.0%, *w*/*w*). The solutions were stirred for 20 min and mixed with glycerol (2.0%, *w*/*v*), heated in a water bath at 60 °C for 30 min and cast onto glass plates (diameter 12.5 cm). Then, the glass plates containing samples were put into the oven to dry (60 °C, 3 h). After drying, the films were peeled off and equilibrated at 25 °C ± 2 °C and 52% ± 2% relative humidity (RH) for at least 48 h before further analyses.

### 2.3. Physicochemical Characterization of the Composite Films

#### 2.3.1. Moisture content (MC), Swelling Capacity (SC), Water Solubility (WS) and Water Vapor Permeability (WVP)

Square films (2 cm × 2 cm, weight as *m*_0_) were dried to a constant weight in a laboratory oven at 105 °C for 24 h and weighed as *m*_1_. The SC of the films was determined by immersing them in 40 mL deionized water for 24 h at ambient temperature. The above-mentioned samples were clamped out. Subsequently, the remaining water on the surface was absorbed by filter paper. Then, the films were weighed as *m*_2_. Afterward, the films were dried at 105 °C for another 24 h, and the weight was recorded as *m*_3_. Therefore, the MC, SC and WS were calculated using Equations (1), (2) and (3), respectively.
(1)$$MC\left(\%\right)=\frac{m_{0}-m_{1}}{m_{0}}\times 100\%$$
(2)$$SC\left(\%\right)=\frac{m_{3}-m_{1}}{m_{1}}\times 100\%$$
(3)$$WS\left(\%\right)=\frac{m_{1}-m_{2}}{m_{1}}\times 100\%$$

WVP was examined using the gravimetric method, as described by Wang, Liu, Ye, Wang and Li [20], with some modifications. In the experiment of this study, a paraffin seal was employed between the composite film and the mouth of a suitable testing cup, which contained 5 g of anhydrous calcium chloride. Subsequently, the testing cups were transferred into a desiccator (75% RH), supplemented with the saturated sodium chloride solution and kept for 24 h. The WVP of the composite film was examined at 25 °C and expressed as Equation 4.
(4)$$WVP=\frac{\Delta w\times L}{t\times A\times \Delta p}$$
where *t* denotes the time (s); *A* represents the measuring area (m^2^); Δ*p* expresses the water vapor pressure difference across the two sides of the film; Δ*w* is the weight change of the cup (g); and *L* expresses the thickness of the film (m).

#### 2.3.2. Light Permeability

*L**, *a**, *b** and Δ*E* were assessed using a colorimeter (CS-800, 400–700n, Hangzhou, China) to determine the color of the composite films. Parameters were recorded randomly for six points selected on the surfaces of each film. 

Light transmittance and opacity were examined using a UV–visible spectrometer (T6, UV/Vis spectrometer, PG Instruments Ltd., Shanghai, China). Films were cut into 10 mm × 30 mm rectangles and placed on a quartz cell for determination. The respective measurement was repeated three times, and the average was taken. The opacity and transparency (T) of the composite films were calculated as follows:(5)$$Opcity=\frac{A_{600}}{d}$$
where *A*_600_ denotes the absorbance value of the composite film at 600 nm; *d* represents the thickness of the composite film (mm).
(6)$$T\left(\%\right)=0.1^{A}\times 100\%$$
where *T* represents the transparency of the composite film; *A* represents the absorbance of the composite film.

### 2.4. Structural Properties of the Composite Films

#### 2.4.1. SEM

Cross-sections of the composite films were observed using a scanning electron microscope (SEM) (JEC 3000FC, JEOL, Tokyo, Japan). Before observation, samples were immersed in liquid nitrogen, cryo-fractured and gold-plated. Afterward, they were transferred to the SEM chamber and observed at an acceleration voltage of 20 kV.

#### 2.4.2. FTIR

The chain interactions into films were analyzed through FTIR spectroscopy with an IS50 spectrometer (ThermoNicolet, Madison, WI, USA). The spectra were collected over a wavenumber range of 4000–400 cm^−1^ and then analyzed using OMNIC Spectra software (Bruker Optics, Champs-sur-Marne, France).

### 2.5. Antibacterial Potential of the Composite Films

Antimicrobial activities of the composite films were assessed with a reported liquid culture assay. *Escherichia coli* (*E. coli*) and *Staphylococcus aureus* (*S. aureus*) were considered the main foodborne pathogens in this study. The LB liquid medium was prepared and placed in an autoclave for 20 min at 121 °C. Subsequently, the container was taken out and cooled down till the temperature was suitable for the medium to be used. The liquid incubation test was performed by cutting the films into squares (2 × 2 cm^2^). Films were immersed into 20 mL nutrient broth supplemented with 100 µL culture of bacteria (10^5^ CFU/mL) and then incubated at 37 °C. During the incubation, the samples (1 mL) were extracted periodically at 24 h and 48 h. Their antibacterial properties were examined by testing the optical densities using a spectrophotometer (T6, 325–1100n, Beijing, China) at 560 nm.

### 2.6. Degradation of the Composite Films

The prepared films were cut into squares (5 cm × 5 cm) and then buried in natural soil at a depth of 5.0 cm to determine their biodegradability. Subsequently, ordinary polyethylene films were also prepared to compare with the composite films as the control. The composite films and the polyethylene films were dug out and then recorded in pictures to assess their biodegradability on the 2nd day, 4th day, 6th day, 8th day, 10th day and 12th day.

### 2.7. Application of the Composite Films

Fresh pork was cut into 30 g and wrapped up in the films (with 1.2 wt% Lys as an example) and then stored in a refrigerator at 4 °C to confirm the antibacterial properties of the composite films. Meanwhile, food-grade polyethylene plastic bags served as the control group to determine the freshness of the meat. During the storage of the chilled pork, images were captured to record changes over 7 days. 

### 2.8. Statistical Analysis

Data are expressed as mean value ± standard deviation of three independent experiments unless mentioned otherwise. The significance level of the differences between means was assessed at *p* < 0.05. SPSS software version 20 was used to analyze the results using a one-way analysis of variance (ANOVA) coupled with Tukey’s test and reported with Origin 2019b.

## 3. Results and Discussion

### 3.1. WS, MC, SC and WVP of the Composite Film

Water solubility (WS), moisture content (MC), swelling capacity (SC) and water vapor permeability (WVP) are the key indicators of water sensitivity of protein-based films and can be used to measure their water permeability. The rapid loss of moisture in low-humidity environments can reduce the mechanical properties of the film, leading to film breakage or shrinkage. As a non-toxic and edible plasticizer, glycerol plays a role in promoting water retention and reducing water loss in the preparation of protein-based composite films. Therefore, the EWP film with and without Lys was prepared with glycerol as the plasticizer, and WS, MC, SC and WVP are shown in Table 1. As can be seen, the WS of the composite film gradually decreased with the increase in Lys addition, from 36.68% ± 2.78% to 26.45% ± 3.15%. The decrease in WS may be caused by the direct interaction between the weakly polar molecules of the EWP and Lys [21]. In addition, SC also showed a downward trend from 4.23% ± 0.36% to 2.84% ± 0.16% (*p* < 0.05). Meanwhile, molecules in the composite films will bind more tightly, which can also reduce the free volume of the composite film matrix, leading to a further decrease in SC. In addition, MC decreased from 12.11% ± 0.79% to 6.33% ± 0.05%. This was due to the increased proportion of Lys in the composite film and the enhanced interaction between proteins and water molecules, resulting in a decrease in the moisture content free in the film substrate and a decrease in MC value [22]. Thus, the addition of Lys has a significant impact on the water permeability of the composite film.

Water vapor permeability (WVP) is another indicator that reflects the water permeability of the composite films [23,24]. It is reported that packaging materials with low WVP can inhibit water loss and reduce nutritional loss while maintaining the complete appearance of the food itself to extend its storage time [25,26]. In general, the WVP value of packaging films is affected by the properties of composite films, film structure and environmental conditions. As can be seen in Table 1, the WVP value of the composite film decreased from the initial value of 14.87 ± 1.78 (gPa^−1^s^−1^m^−1^·10^−10^) to 9.30 ± 0.66 (gPa^−1^s^−1^m^−1^·10^−10^) with the addition of Lys, showing a significantly decreasing tendency of WVP of the composite film. This phenomenon was due to the increase in cross-linking reactions between Lys and other substances in the film, such as proteins and glycerol [27,28]. Furthermore, the transfer path of water molecules through the membrane substrate was more difficult after the addition of Lys [29]. 

### 3.2. Light Permeability of the Composite Film

The color of the EWP-Lys composite film is explained in Figure 1a. There were no significant changes in L* values and ΔE values with the increasing concentration of Lys. Overall, the b* value did not increase significantly, while the a* value increased from −0.75 ± 0.16 to −1.26 ± 0.04, indicating that the greenness value of the composite films increased significantly when the Lys addition reached 2%. The influence of Lys on the light performance of composite films is also manifested in its impact on optical properties. Similarly to the changes in color differences, it can be seen from Figure 1b that the concentration of Lys has no significant impact on the opacity and light transmittance of the EWP-based composite film (*p* > 0.05). However, the opacity of the composite film presented a downward trend, and the light transmittance increased when the amount of Lys added reached 1.2%. This could be attributed to the increase in protein–protein cross-linking by the participation of Lys in the film-forming reaction, which increased the molecular matrix space and then caused an increase in the visible light transmittance [30]. However, the addition of Lys had no significant impact on the optical properties of the composite films from the overall trend.

### 3.3. Structural Properties

#### 3.3.1. SEM Image Analysis

It has been reported that Lys, whose essence is protein, could be embedded into the macromolecular chain of EWP through protein–protein interactions, promoting the fluidity of the molecular chain and reducing the brittleness of EWP-based films [31]. Figure 2a–f represents scanning electron microscope images of EWP-Lys composite film with a concentration of Lys from 0% to 2.0% (*w*/*w*). It can be observed from Figure 2 that significant differences appeared in the cross-sectional structure and morphology of the composite films. With an increasing addition of Lys, the cross-section of the composite films gradually changed from “convolution” to “smoothness”, which could be interpreted as the decrease in intermolecular force between proteins through hydrogen bonds between Lys and EWP under high-speed rotation and relatively high temperature conditions [32,33]. In addition, with the increase in the amount of Lys, the degree of cross-linking between Lys and EWP increased, and the degree of convolution on the protein-based film further decreased. At the same time, it increased the tortuosity of water molecular inlet and outlet channels, which was one of the reasons why WVP decreased with the increase in Lys. It was found by Lan, Liu, Wang, Tian, Miao, Wang and Tang [34] that the morphology of lysine/starch/PVA film was denser than that of starch/PVA film. The hydroxyl groups of starch can form hydrogen bonds with the amino groups of Lys, forming a dense structure.

#### 3.3.2. Fourier-Transform Infrared Spectroscopy Analysis

Intermolecular interactions and changes in protein structure within the film substrate can be studied at the molecular level. As reported, molecules of Lys have hydrophilic groups (amino acid residues) that can interact with the matrix components of EWP films [35,36]. To clarify the intermolecular interactions between Lys and EWP, FTIR spectra of EWP films prepared with glycerol and different percentages of Lys are depicted in Figure 3. The characteristic peaks of EWP mainly occurred at 3404 cm^−1^, 2945 cm^−1^, 1639 cm^−1^, 1537 cm^−1^ and 1043 cm^−1^, attributed to O–H and N–H stretching, C–H stretching, C=O stretching, N–H bending and C–O–C stretching vibrations, respectively [37,38,39]. After combining with Lys, no additional peaks of the films were observed, indicating that there was no covalent bond between EWP and Lys [40]. In other words, the interaction between the compounds is more likely due to physical reactions, and the degree of cross-linking between Lys and EWP is high enough. However, the O–H stretching vibration was depicted from 3379 cm^−1^ to 3404 cm^−1^ with the concentration of Lys increasing from 0% (*w*/*w*) to 2.0% (*w*/*w*), which might be attributed to the alteration of the stretching vibration in free O–H bands. In addition, C=O stretching was detected at 1662 cm^−1^ and slightly shifted to 1639 cm^−1^, while C–N stretching and N–H bending were observed shifting from 1539 cm^−1^ to 1537 cm^−1^ with the increasing addition of Lys. These phenomena were caused by C=O stretching and N–H bending in Lys in the film matrix [41]. Therefore, the spectral changes of EWP-based films should be due to alterations in the orientation and conformation of peptide chains with the incorporation of Lys.

### 3.4. Antibacterial Properties

Figure 4 shows the growth of *Escherichia coli* (*E. coli*, Figure 4a) and *Staphylococcus aureus* (*Sau.*, Figure 4b) at 24 h and 48 h (represented by changes in absorbance values) after adding the antibacterial composite film with different concentrations of Lys to the liquid medium. It can be seen from Figure 4 that when the concentration of Lys reaches 2.0%, the absorbance is the lowest, and the bacteriostatic effect is the best. In addition, a stronger inhibitory effect on *E. coli* was exhibited in Figure 4a, supporting the broad-spectrum antibacterial activity of Lys [42]. The addition of Lys indeed improves the antibacterial activity of the composite film and also proves that the EWP-Lys film has a good antibacterial application prospect. The possible bacteriostatic mechanism of Lys lies in the destruction of the structure of the cell membrane of microorganisms, leading to the separation of cell walls and cytoplasm, cell collapse or cytoplasmic overflow [43,44]. Secondly, the normal physiological metabolism of bacteria could be disrupted, causing disruptions in cell material, energy and information transmission [45,46]. Thirdly, Lys can interact with internal substances in cells to destroy the cell core, resulting in the overflow of ultraviolet absorbents and ultimately leading to the death of the cell [47]. However, in-depth studies should be conducted for more information to determine the exact bacteriostatic mechanism of the composite films.

### 3.5. Biodegradability

The biodegradability of the EWP-Lys composite film with the traditional plastic (polyethylene, PE) film as a control group is shown in Figure 5. Both groups were buried in the soil for several days, and the degradation within 12 days was observed and compared. The results showed that no signs of degradation could be seen within 12 days for the control group, while the EWP-Lys composite film exhibited degradation phenomena starting from the 4th day with wrinkles and structural defects. As time increased, the EWP-Lys group gradually fractured and lost and degraded into fragments of the size of soybean seeds by the 12th day. However, the size and morphology of the control group did not change at all. Therefore, the biodegradability of EWP-Lys films is high enough, and it is estimated that it will take about two weeks for completely degradation. The excellent degradation of EWP-Lys composite film provides great potential for its subsequent practical application, without any concern about environmental damage [48].

### 3.6. Application

#### 3.6.1. pH Values of the Chilled Pork during Storage

Meat products are prone to microbial contamination during storage due to their rich protein content. Bacterial microorganisms are extremely prone to decomposing proteins into ammonia substances, resulting in an increase in pH [49]. Figure 6 shows the change in pH of the chilled pork during its storage period when wrapped in the composite film (with 1.2 wt% Lys), while the chilled pork wrapped in PE film was used as the control group. As shown, the pH of the Lys group on the first and seventh days of storage is slightly higher than that of the control group. This is because a small portion of the protein present in the composite film might adhere to the chilled pork, and the protein is oxidized and decomposed to release alkaline groups at the beginning of storage, resulting in a higher pH than that of the control group. On the seventh day, the water molecular channel of the composite film is opened due to protein decomposition, resulting in an increase in the pH in the environment. Considering the actual storage situation, when this composite film is applied to the preservation of cold fresh meat, the recommended storage time is 3–5 days, and the preservation effect is slightly higher than that of ordinary polyethylene film.

#### 3.6.2. Color of the Chilled Pork during Storage

The color of chilled meat is the most intuitive method used by consumers to evaluate the freshness of meat products [50]. The higher the redness value (a*) of displayed meat, the more popular it is with consumers. Therefore, the a* value is a very important indicator in the pork color evaluation system. Table 2 shows the impact of two types of packaging (i.e., PE film and EWP-Lys film) on the color value of chilled pork. As can be seen, the brightness value (L*) of chilled pork shows an upward trend with the extension of storage time, while the yellowness value (b*) is not significantly affected during the whole storage period. The redness value (a*) shows a downward trend with the prolongation of storage time, mainly due to the gradual increase in the degree of lipid oxidation of the chilled pork and the color change caused by water loss over time. In the early stage of storage, the a* value of chilled pork in different packages changed slowly. After storing for 3 days, the a* value of the composite film increased and was significantly lower than that of the control group (PE film) (*p* < 0.05), indicating that the storage water loss rate of the chilled pork wrapped in PE film was greater than that of the experimental group. Thus, the composite film has a good preservation effect in practical applications.

#### 3.6.3. Total Bacterial Counts of the Chilled Pork during Storage

Cold fresh meat is susceptible to microbial contamination during storage [51]. Figure 7 shows the growth of microorganisms in chilled pork wrapped in PE film and EWP-Lys composite film during the storage period. It can be clearly seen from the figure that there is a slight growth of microorganisms in the Lys group at the initial stage of storage, which is consistent with the measurement results of indicators such as the pH of chilled pork. With the increase in storage time, the microbial growth of the Lys group was significantly lower than that of the control group (PE film) after 3 days (Table 3). Lys in the composite films played an effective role in inhibiting microbial growth, which means that the bacteriostatic effect of Lys was initially effective. After the storage time was extended to 7 days, the microbial growth rate was still lower than that of the control group. This result can more intuitively demonstrate that the composite film can play a key role in the storage process of cold fresh meat.

## 4. Conclusions

Lys is used as a bacteriostatic agent with different concentrations in EWP-based composite films to improve their physico-chemical and antibacterial properties, aiming to develop alternative and environmentally friendly composite films applicable to the food-packaging field in chilled meat. Measurements of biodegradation seemed to support the idea that the composite films were much easier to degrade than the polyethylene films, which demonstrated excellent biodegradability. Their structural properties showed that a stronger cross-linking occurred with the increasing addition of Lys. Furthermore, antibacterial effects on microorganisms have been proved and demonstrated in the application of chilled pork. Thus, the current study demonstrates that EWP-Lys composite films could be considered as an ideal option for environment-friendly food packaging, especially for chilled meat, due to the improved physico-chemical properties, antibacterial properties and degradable properties.

## Figures and Tables

**Figure 1 foods-12-02431-f001:**
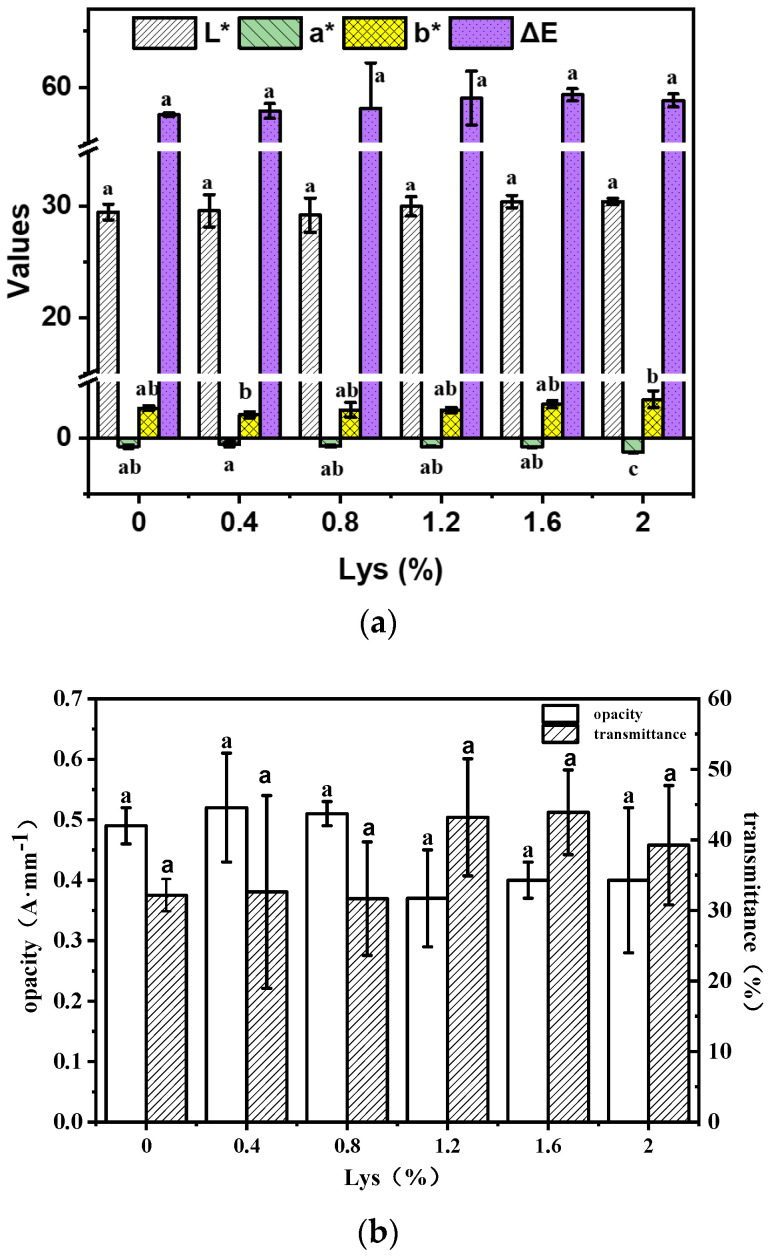
(**a**) Color parameters and (**b**) opacity and transmittance of antibacterial composite films incorporated with different concentrations of Lys (0–2.0%). Different letters indicate significant differences (*p* < 0.05).

**Figure 2 foods-12-02431-f002:**
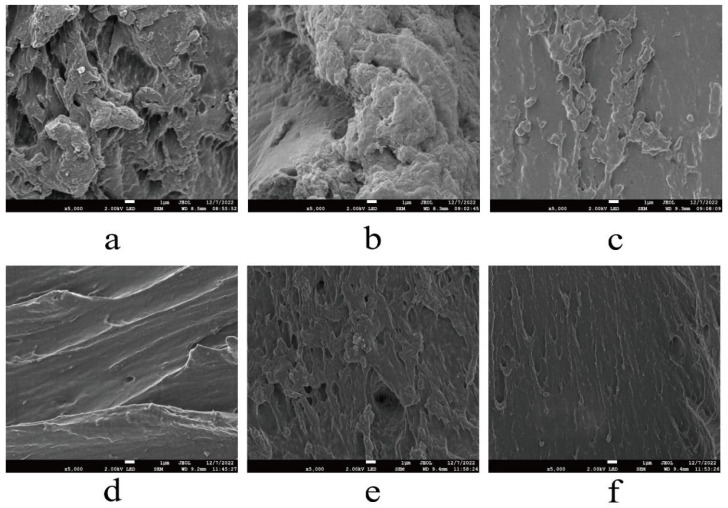
Effects of different concentrations of Lys ((**a**–**f**), 0–2%) on the cross-section structure of composite films.

**Figure 3 foods-12-02431-f003:**
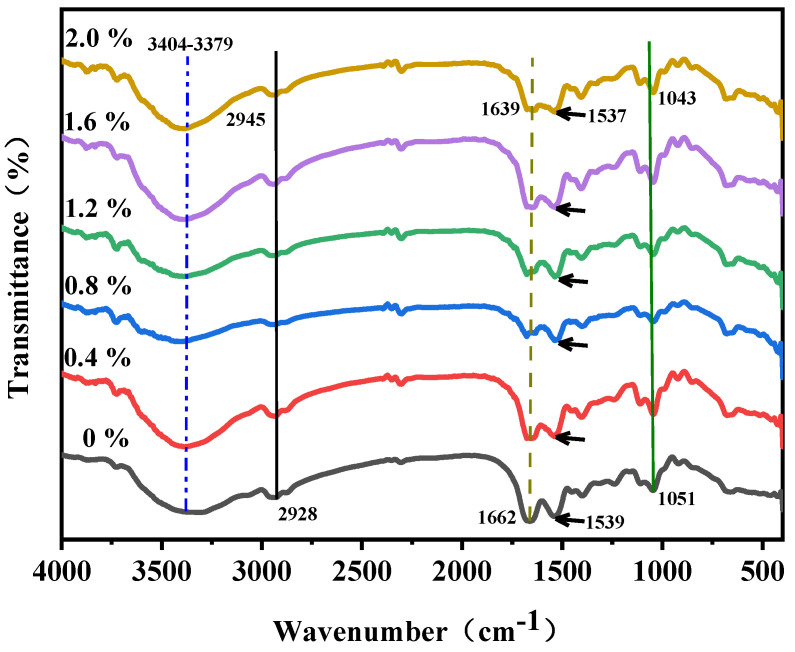
Infrared spectroscopy (FTIR) analysis of composite films with different concentrations of Lys (the black line: 0% Lys, the red line: 0.4% Lys, the blue line: 0.8% Lys, the green line: 1.2% Lys, the purple line: 1.6% Lys, the yellow line: 2.0% Lys).

**Figure 4 foods-12-02431-f004:**
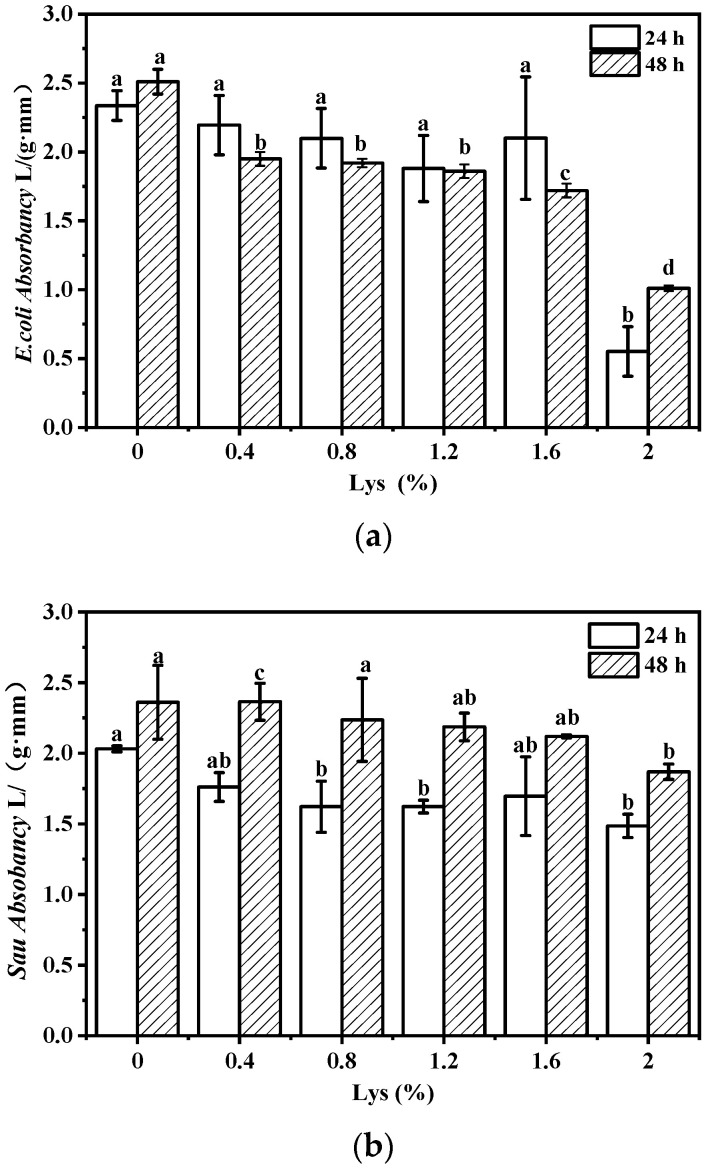
The antibacterial effects of antibacterial films on *Escherichia coli* (*E. coli*) (**a**) and *Staphylococcus aureus* (*Sau.*) (**b**) within 24 h and 48 h. Different letters indicate significant differences (*p* < 0.05).

**Figure 5 foods-12-02431-f005:**
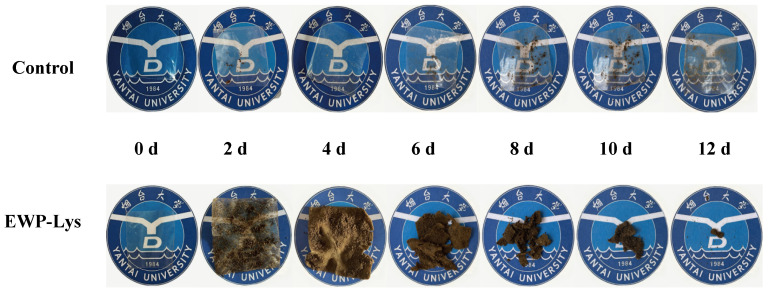
Comparison of degradation effects of polyethylene plastic film (control) and the composite film (EWP-Lys) with 1.2% Lys.

**Figure 6 foods-12-02431-f006:**
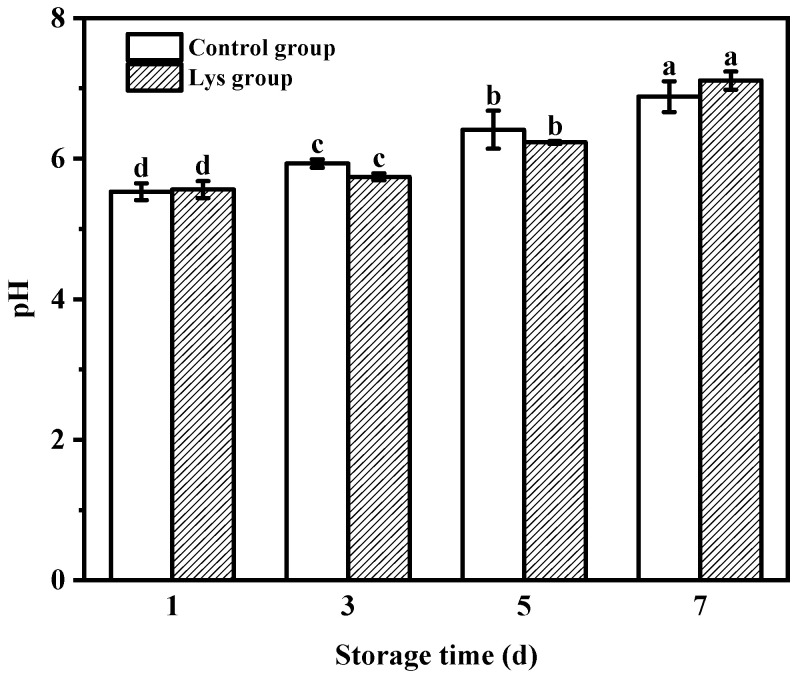
Changes in the pH value of the chilled meat wrapped in polyethylene plastic film (control group) and the composite film (Lys group, with 1.2 wt% Lys) during storage (mean ± standard error). Different letters indicate significant differences (*p* < 0.05).

**Figure 7 foods-12-02431-f007:**
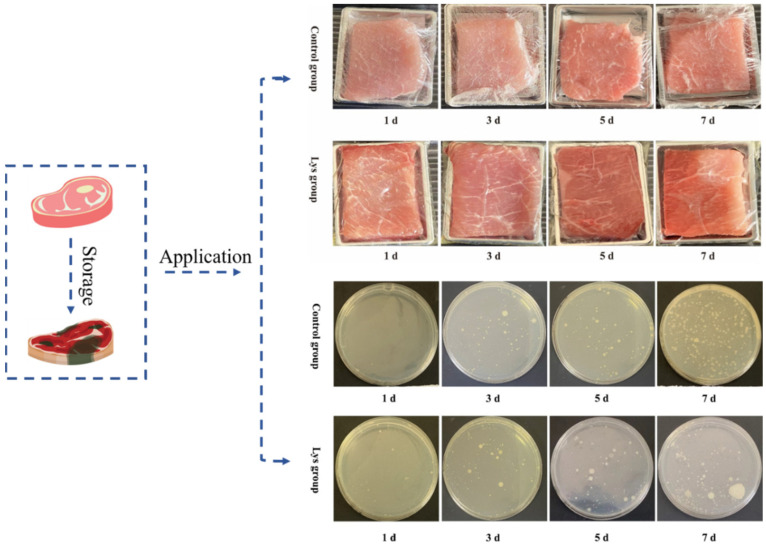
The application of the EWP-Lys films (with 1.2 wt% Lys as an example) in the preservation of chilled pork, including the apparent images and the colony changes (polyethylene film was used as a control group).

**Table 1 foods-12-02431-t001:** Water solubility (WS), moisture content (MC), swelling capacity (SC) and water vapor permeability (WVP) of the composite films with different concentrations of Lys.

Lys (%)	WS (%)	MC (%)	SC (%)	WVP (gPa^−1^s^−1^m^−1^·10^−10^)
0	36.68 ± 2.78 ^a^	16.34 ± 2.13 ^a^	4.23 ± 0.36 ^a^	14.87 ± 1.78 ^a^
0.4	35.78 ± 1.51 ^a^	13.22 ± 1.39 ^ab^	4.15 ± 0.10 ^ab^	14.73 ± 0.40 ^a^
0.8	33.99 ± 1.19 ^ab^	14.39 ± 1.21 ^ab^	3.72 ± 0.19 ^bc^	14.37 ± 0.93 ^a^
1.2	33.03 ± 3.62 ^ab^	12.94 ± 2.29 ^ab^	3.63 ± 0.26 ^c^	13.53 ± 0.06 ^a^
1.6	29.77 ± 0.33 ^bc^	15.21 ± 1.34 ^a^	3.19 ± 0.08 ^d^	11.60 ± 0.66 ^b^
2.0	26.45 ± 3.15 ^c^	10.27 ± 1.29 ^b^	2.84 ± 0.16 ^d^	9.30 ± 0.66 ^c^

Values are presented as mean ± standard deviation. Different letters in the same column indicate significant differences (*p* < 0.05).

**Table 2 foods-12-02431-t002:** Changes in the Chrominance value of the chilled meat during storage (mean ± standard error).

Chromatic Aberration	Storage Time (d)	Control Group	Lys Group
L*	1	55.69 ± 1.11 ^c^	42.16 ± 0.79 ^b^
	3	59.35 ± 0.54 ^b^	41.03 ± 1.89 ^b^
	5	61.71 ± 1.06 ^a^	46.37 ± 0.50 ^a^
	7	61.12 ± 0.06 ^a^	46.54 ± 2.92 ^a^
a*	1	3.07 ± 0.40 ^a^	2.07 ± 0.34 ^b^
	3	2.05 ± 0.30 ^b^	3.47 ± 0.41 ^a^
	5	0.89 ± 0.08 ^c^	1.87 ± 0.03 ^b^
	7	0.48 ± 0.04 ^d^	1.46 ± 0.06 ^c^
b*	1	12.82 ± 0.45 ^a^	10.57 ± 0.85 ^a^
	3	8.06 ± 0.82 ^c^	9.86 ± 1.32 ^a^
	5	11.58 ± 0.61 ^b^	8.39 ± 0.09 ^b^
	7	10.90 ± 0.04 ^b^	10.33 ± 0.44 ^a^
Δ E	1	34.68 ± 1.03 ^a^	47.38 ± 0.65 ^a^
	3	30.49 ± 0.54 ^b^	48.95 ± 1.80 ^a^
	5	29.46 ± 0.86 ^b^	43.79 ± 0.17 ^b^
	7	27.48 ± 1.89 ^c^	41.04 ± 3.95 ^b^

Values are presented as mean ± standard deviation. Different letters in the same column indicate significant differences (*p* < 0.05).

**Table 3 foods-12-02431-t003:** The number of colonies in the chilled meat during storage (mean ± standard error).

Group	1 d (CFU)	3 d (CFU)	5 d (CFU)	7 d (CFU)
Control	(1.00 ± 0.28) × 10^−4^	(8.15 ± 1.63) × 10^−4^	(1.82 ± 0.37) × 10^−5^	(3.85 ± 0.27) × 10^−5^
Lys	(2.85 ± 0.35) × 10^−4^	(4.90 ± 0.56) × 10^−4^	(9.85 ± 0.92) × 10^−4^	(1.57 ± 0.30) × 10^−5^

Values are presented as mean ± standard deviation.

## Data Availability

The datasets generated for this study are available on request to the corresponding author.

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
