# Peer review of "Characterization and Antibacterial Properties of Egg White Protein Films Loaded with ε-Polylysine: Evaluation of Their Degradability and Application"

_foods, 2023, doi:10.3390/foods12122431_

Round 1

Reviewer 1 Report

Dear authors, thank you for your interesting article. 

I recommend you correcting a number of comments below. 

1. Different abbreviations for ε-polylysine are used in the text of the manuscript, in the names of the tables and in the figures: PLYs, Lys, PLys, lysine. If the authors mean the same substance, one abbreviation should be used everywhere.

2. Similarly, choose a designation for swelling capacity: SC or SR.

3. It is not clear why polylysine concentrations are chosen: from 0.4%, to 2.0%. Have the properties of films at higher concentrations been studied, e.g. 5%, 10%?

4. In paragraph 3.6.1 the authors conclude (Considering the actual storage situation, when this composite film is applied to the preservation of cold fresh meat, the recommended storage time is 3-5 days and the preservation effect is slightly higher than that of ordinary polyethylene film.) However, in figure 6 it is seen that for days 1,5,7 the differences in pH are not significant as compared with controls. Also, the authors explicitly indicate the release and decomposition of protein on the surface of the meat for the composite film under study, which does not occur for the control, and this is obviously a disadvantage of the composite. Please comment.

5. For the credibility of the experiment presented in the figure , it is necessary to extend the description of the research process in point 2.7, and to demonstrate not only the pictures of the petri dishes, but also the quantitative data (number of colonies).

6. Why was the film with a Lys concentration of 1.2% chosen for the biodegradation study? Were other composites investigated?

7. The same question about Section 3.6. It should be clearly stated what concentration of Lys was used to create the composite in these experiments. In addition, make appropriate conclusions in the Conclusions section.

Reviewer 2 Report

1         I suggest changing all the keywords, as they are all present in the title of the article, they need to be different, this impacts the search for your article

2         In the middle of line 42, it would be appropriate to insert explanations of the application of egg white protein as a polymer, it was even mentioned that it has the ability to form gel, but nothing that relates to an application as a polymer.

3         In the introduction you talk about the importance of combining egg white protein with other polymers due to their low properties. However, that's not what you did, you added a compound with antimicrobial capacity only. I believe that the introduction should be completely reorganized. You cite some studies on the addition of polymers to EWP, such as carrageenan and chitosan, so I understand that you are going to do something in this direction as well. The introduction needs to be completely improved, highlighting the main points that can be remedied, with your research, something along those lines.

4         Other chemicals used in this study were of analytical grade?? And why were they not mentioned? either quote them all, or remove that sentence

5         Item 2.2 When was glycerol added?

6         How long did the WVP analysis take place?

7         In item 2.7, was the meat contaminated, or does it only have the natural microbiota?

8         Lines 161-163, the sentence is confusing, I don't understand the objective. Only from protein films? Water permeability is a separate analysis, we do not use WS, MC and SR values ​​to determine it

9         Item 3.2 (results), why are these properties important? what do they provide to the food being packaged? it is interesting to add a little explanation about the influence of the color and opacity of a package on the packaged food

10     Perhaps it would be more interesting to add WVP results in table 1 and remove figure 1a

11     Item 3.4 What explanation do you attribute to the 24h time to present an absorbance lower than the 48h time? Does the film have an inhibitory action only in 24 hours and then it starts to lose its action? to what extent is this desirable?

12     Escherichia coli and Staphylococcus aureus should be in italics. Review the entire document

13     Was the pH measured that of the film or that of the meat? Line 311-312 got confused, about that

14     lines 350-361, don't you have a colony count? It would be more interesting to present the numbers as well

15     Line 371 suggest removing "highly" attractive

You can check, and give an improvement

Round 2

Reviewer 2 Report

The authors endeavored to reorganize the work, taking into account all the highlighted points. In my view the work has been improved.

one last point (it is interesting that the keywords are smaller) I suggest: Bioactive film, ecologically friendly, application in situ. Something along those lines